# Segmentation of Unsound Wheat Kernels Based on Improved Mask RCNN

**DOI:** 10.3390/s23073379

**Published:** 2023-03-23

**Authors:** Ran Shen, Tong Zhen, Zhihui Li

**Affiliations:** 1College of Information Science and Engineering, Henan University of Technology, Zhengzhou 450001, China; 2Key Laboratory of Grain Information Processing and Control, Henan University of Technology, Ministry of Education, Zhengzhou 450001, China

**Keywords:** wheat, mask RCNN, target segmentation, feature pyramid network, attention mechanism

## Abstract

The grade of wheat quality depends on the proportion of unsound kernels. Therefore, the rapid detection of unsound wheat kernels is important for wheat rating and evaluation. However, in practice, unsound kernels are hand-picked, which makes the process time-consuming and inefficient. Meanwhile, methods based on traditional image processing cannot divide adherent particles well. To solve the above problems, this paper proposed an unsound wheat kernel recognition algorithm based on an improved mask RCNN. First, we changed the feature pyramid network (FPN) to a bottom-up pyramid network to strengthen the low-level information. Then, an attention mechanism (AM) module was added between the feature extraction network and the pyramid network to improve the detection accuracy for small targets. Finally, the regional proposal network (RPN) was optimized to improve the prediction performance. Experiments showed that the improved mask RCNN algorithm could identify the unsound kernels more quickly and accurately while handling adhesion problems well. The precision and recall were 86% and 91%, respectively, and the inference time on the test set with about 200 targets for each image was 7.83 s. Additionally, we compared the improved model with other existing segmentation models, and experiments showed that our model achieved higher accuracy and performance than the other models, laying the foundation for wheat grading.

## 1. Introduction

As one of the three staple foods for humans, the quality of wheat affects national food safety, agricultural production development, and people’s living standards [1]. Grains that are damaged but still useful are called unsound kernels, including injured, spotted, broken, sprouting, and moldy kernels. The proportion of unsound kernels is an important reference for measuring wheat quality. Therefore, in the flow of wheat, realizing the rapid, accurate, and non-destructive detection of unsound kernels is of great significance for the assessment of wheat quality.

Currently, there are five mainstream detection technologies for unsound kernels. Below, we enumerate the papers addressing each technology and summarize the current shortcomings.

The first technology is manual detection, which refers to picking and classifying the extracted samples one by one relying on human eyes and experience [2]. On the one hand, it is time-consuming and labor-intensive. On the other hand, the detection accuracy is low because of subjective judgments.

The second technology is the use of the spectral information of unsound kernels for detection. Spectral features, image features, and fused spectral and image features of hyperspectral images of unsound wheat grains were extracted by [3] and used as the input of a support vector machine (SVM) [4]. The results showed that the fused spectral and image information was most conducive to classification. In [5], hyperspectral images and high-resolution images of unsound wheat kernels were registered and fused as the input of a VGG [6] network model. The average precision of the fused image was higher than that of the individual hyperspectral image and individual high-resolution image. Although this method could extract more information from images, the equipment for collecting spectral information is expensive.

The third technique is based on acoustic information for detection. Pearson [7] used shock acoustics to detect worm-eaten kernels. The worm-eaten kernels were placed on a steel plate, and acoustic signals were generated when the particles hit the steel plate. Linear discriminant analysis (LDA) [8] was used for feature extraction and classification, and the recognition accuracy rate was 84.4%. However, the method based on acoustic wave information relies too much on the sound propagation medium. Furthermore, it is greatly affected by noise, resulting in low detection accuracy.

The fourth technique is the use of traditional machine learning methods for detection. First, the features of the unsound wheat kernels are extracted manually, and then classifiers such as SVM and principal component analysis (PCA) [9] are used for classification. However, features obtained through manual extraction are complex and inaccurate. Furthermore, this technique is ineffective for the segmentation of sticky kernels.

The fifth technique uses deep learning methods for detection. Many researchers [10,11,12] have proposed various optimization algorithms for classification. However, these studies focused on simple classification tasks, without exploring further instances of segmentation algorithms. Furthermore, the datasets were artificially placed in advance, so it was difficult to satisfy the practical application requirements. At the same time, traditional segmentation methods are often used for classification tasks. For example, Shatadal [13] achieved segmentation by corroding and expanding grains and then filling cavity areas. Siriwongkul [14] obtained the separation end points based on the edge to draw the dividing line and achieve segmentation. However, the results of traditional segmentation are not good. Moreover, most semantic segmentation methods are invalid for wheat kernels with a dense distribution and a high degree of adhesion. This is because the algorithm recognizes different parts of a single target separately during the classification, which leads to multiple recognition results for one target. Take the famous U-Net algorithm as an example, whose segmentation results are shown in Figure 1. It can be seen that the semantic segmentation produced a poor result, even for wheat kernels with a low adhesion degree.

Although the mask RCNN model is widely used in various fields because of its good performance and accuracy for large objects and objects with sparse distribution, the accuracy is relatively low in the fine-grained segmentation of small objects, such as unsound kernels with a dense distribution. The reason for this is that the characteristics of wheat grains, such as their oval shape and yellow-white skin color, are similar, which makes feature extraction networks less effective.

This paper proposes an instance segmentation algorithm for unsound wheat kernels based on an improved mask RCNN to address the aforementioned problems. On the one hand, it solves the problem of adhesion between densely distributed targets and realizes fast, accurate, and non-destructive detection, laying the foundation for the grading of wheat. On the other hand, the accuracy is higher, which provides ideas and inspiration for the application of mask RCNN in the fine-grained segmentation. The basic structure of the article is as follows: Section 2 describes the dataset acquisition process, the improved mask RCNN network, and the evaluation metrics for object segmentation. Section 3 evaluates the model. Section 4 discusses the results, and Section 5 provides a summary and a future outlook.

## 2. Materials and Methods

### 2.1. Experimental Settings

#### 2.1.1. Datasets

Unsound kernels refer to wheat grains that are damaged but still useful, including injured kernels, spotted kernels, broken kernels, sprouted kernels, and moldy kernels. Injured kernels refer to grains that are eroded by insects, which will cause damage to the embryo or endosperm. The disease spot and a dark brown or black embryo are visible on the surface of the spotted kernels. Additionally, wrinkled kernels with a white, purple, or pink mold surface, and black ascus shells, are also spotted kernels. Broken kernels are usually flattened and broken. Gray mold usually adheres to the epidermis of moldy kernels. If the bud or young root breaks through the seed coat and does not exceed the length of the kernel, it is a sprouted kernel. If the bud or young root does not break through the seed coat, but the germ coat has ruptured and separated from the embryo, you can also call it a sprouted kernel. If there are no characteristics of the above types, we call them perfect kernels. Figure 2 shows the appearance of different kinds of wheat kernels. 

One hundred and thirty images were collected and labeled using the image annotation software (https://github.com/wkentaro/labelme, accessed on 10 May 2022) to obtain the annotated mask image. For kernels covered by others, the covered part is not labeled, as is shown in Figure 3. To improve the recognition accuracy and purposefully emphasize the local characteristics and region of interest, we used five data enhancement methods to process images, including reducing brightness, increasing noise, adding random points, translating, and flipping. Finally, we had a total of 780 images that we divided in the ratio of training set to validation set to test set = 8:1:1. That is to say, we obtained the 624 training set, the 78 validation set, and the 78 test set, with specific statistics listed in Table 1.

#### 2.1.2. Experiment Equipment 

We used a 900 mm camera holder to build the data acquisition platform and used a Hikvision industrial camera with 10 million pixels, model MV-CE100-30GC, to obtain the datasets. A server with two Tesla T4 graphics cards was used, and the operating system was Ubuntu 18.04, 64-bit. The model was trained on Cuda 10.1, PyTorch 1.10, and Python 3.7. When training, the batch size is 2, the initial learning rate is 0.0025, and the warm-up method is used.

### 2.2. Improved Mask RCNN

#### 2.2.1. Improved Mask RCNN with AM

Attention mechanisms (AM) are a way to achieve network-adaptive attention. It enables the network to pay attention to what it needs to do, which can be divided into channel attention (CA) [15], spatial attention (SA) [16], and a combination of the two. In the process of identifying unsound kernels, different kernels have different channels and spatial information. For example, spotted and moldy kernels focus on the color characteristics of the embryonic part, while injured kernels focus on the texture information of holes in the cortex and endosperm. Therefore, in this paper, the classic ECA [17] was adopted to extract features from the backbone. We obtained the heat-MAP in Figure 4 using Grad-CAM [18], which clearly showed the region of interest for each type of kernel. In addition, when compared to the heat-MAP of the original ResNet [19] backbone, the model with the fusion of attention mechanisms could learn and utilize the information more pertinently.

#### 2.2.2. Improvement in FPN

In convolutional networks, high-level networks often respond to semantic features, while shallow networks respond to low-level features such as target location information [20]. Therefore, the size of future maps in a high-level network is small, and the geometric information is not enough. So, the classification ability is greatly reduced. Although the shallow network has more geometric information, it has few semantic features, which are not conducive to classification either. Thereupon, the information flow path of the last layer of information extracted from the backbone is too long [21]. So, we added a bottom-up fusion path, which does not disturb the backbone, and further reduced the flow path of high-level information to obtain low-level features [22]. The improved part is shown in the blue dashed box in Figure 5. The 3 × 3 convolution operation of the last part of the original FPN was removed, and the four feature layers (P2, P3, P4, and P5 in Figure 5) were directly taken for another fusion. First, P2 was added to P3 to obtain the fused feature N3, and 3 × 3 convolution was performed on the existing N3 to obtain the final predicted feature layers. Similarly, we pooled N3 and N4 and convolved after fusion to finally obtain the four predicted feature layers. Now, the four prediction feature layers obtained more low-level information, which further improved the positioning ability of the model.

#### 2.2.3. Improvement in RPN

The essence of RPN [23] is a classless object detector based on sliding windows, which divides the coordinates of a pixel in the feature map extracted from the backbone into regions according to the number of pixels. Through the anchor mechanism, some candidate boxes of possible targets can be generated in the regions corresponding to the original map of each pixel. The RPN is used to determine whether the regions for each pixel contain targets and extract the suggestion box.

Considering the dense distribution and high similarity of kernels on a single image, to reduce the amount of computation and memory consumption, we solved it in the following three ways: First, the number of anchor boxes [23] is set to 6, the zoom ratio of the anchor box is defined as {8,16,32,64,128,256}, and the aspect ratio representing the ratio of the length and width of each anchor box is defined as {0.5, 1.0, 2.0}. Second, in RPN, when non-maximum suppression (NMS) [24] is used to further filter the proposal boxes, it is necessary to compare the IOU of the two boxes with the specified threshold. Additionally, if it is higher than our threshold, it means that they identify the same object and then remove boxes that have a lower score. The threshold is reduced to avoid the model identifying the two targets as one whole target. By the way, NMS is the post-processing operation. That is, after getting the regression bounding box, it just selects the best one among the obtained proposals. So the change in threshold will not affect the success rate but will reduce the missed detection rate. Finally, when filtering the foreground and background, it is necessary to calculate the IOU of the proposal box and the ground truth corresponding to the anchor. In order to further reduce the amount of computation, we set the value of the IOU to 0.7, indicating that it is the foreground if the overlap ratio is greater than 0.7, and that it is the background if it is less than 0.3, and the data between the two values will be discarded. We designed experiments to verify the effect of the improvement in RPN. The visualization images before and after anchor box modification are shown in Figure 6a,b. The comparison of the prediction results of the model before and after the improvement in the NMS on adhesion kernels is shown in Figure 6c,d. Moreover, during training, the occupation of video memory was reduced from 14,756 MB to 11,678 MB. During inference, it was reduced from 12,500 MB to 1101 MB. These show that the improvement in RPN is remarkable.

### 2.3. Evaluation Metrics

At present, the academic community usually measures the performance of segmentation algorithms based on three aspects: running time, memory usage, and accuracy [10]. The main index of running time is FPS (frames per second), which refers to the number of pictures that can be inferred by the model per second. In our paper, we used time to evaluate the rate of prediction, which is equal to the reciprocal of the FPS. Memory usage includes parameters. Parameters, which measure the size of model, refer to the total number of parameters of the model. Recall, AR (average recall), precision, AP (average precision), mAP (mean AP), IOU, and MIOU (mean IOU) are measures of accuracy. Recall is the ratio of the number of correctly predicted samples of a certain category to the total number of ground truth samples of this category, as shown in Equation (1). AR is the average of all recalls, which measures whether the positioning of a model checkbox is accurate. Precision is the ratio of the number of correctly predicted samples to the total number of samples, as shown in Equation (2). AP is the area of precision-recall figures. MAP is the average of APs across all categories. IOU is the ratio of the intersection and union of the real value set and the predicted value set of pixels, as shown in Equation (3). MIOU is the class’s average IOU.
(1)$$\mathrm{recall}=\frac{\mathrm{TP}}{\mathrm{TP}+\mathrm{FN}}$$
(2)$$\mathrm{precision}=\frac{\mathrm{TP}}{\mathrm{TP}+\mathrm{FP}}$$
(3)$$\mathrm{IOU}(E,F)=\frac{E\cap F}{E\cup F}$$

In the formula, TP (true positive) is the number of positive samples predicted to be positive samples. TN (true negative) is the number of negative samples predicted as negative samples. FP (false positive) refers to the number of negative samples predicted as positive samples. FN (false negative) is the number of positive samples predicted to be negative. E is the area of the prediction box, and U is the area of the ground truth box.

## 3. Results

### 3.1. Resulting Images

We used a total of 624 training sets and 78 verification sets. The input images are set to a uniform size of 512 × 512, the batch size is 4, the number of iterations is 50, the learning rate is 0.00125, the maximum suppression screening threshold is 0.5, and the foreground and background screening thresholds are 0.7 and 0.3, respectively. We used the same wheat images as the input for the original model and the improved model, getting the prediction shown in Figure 7. The original model will miss targets and pick kernels by mistake, but the improved model can well overcome the problem. Judging from the detection results, the improved model has a better effect. Judging from the detection results, the improved model has a better effect.

### 3.2. Quantitative Results

We extract the feature map of the first four layers in the backbone network and add attention modules to them so that it can focus from all over to the region that it should focus on. In addition, in order to reduce the length of the flow route of high-level characteristic figures and enhance their ability to localize, we add a top-down fusion path in the FPN part to shorten the information flow path of the low-level information. Figure 8 shows the training loss curves and mAP curves of the original model, the model only adding attention mechanisms, the model only modifying FPN, and the model integrating attention mechanism and FPN. It can be seen from the figure that training losses tend to be flat around the 20th epoch. The training loss after adding the attention mechanism and bottom-up FPN is about 3% lower than that of the original model, and the mAP increases by about 3%. The combined attention mechanism and bottom-up FPN model have the lowest loss and the highest mAP value, which shows that the model integrating attention mechanisms and bottom-up FPN is better.

Table 2 contains detailed data on the precision, AP, recall, parameters, and timing of the four models mentioned above. These evaluation indicators were introduced in Section 2.3. From the perspective of horizontal comparison, recall performs better than precision, which indicates that the localization ability of the model is better than the classification ability [25]. From the perspective of a longitudinal comparison, the original model has the lowest prediction accuracy and recall rate. The model that integrates the attention mechanism and bottom-up FPN has the highest accuracy and recall rate. The original model has the lowest memory consumption and the lowest FPS due to the smallest network depth. The model integrated the attention mechanism and bottom-up FPN, which has a larger number of parameters to load and a longer time to predict. However, since we reduced the number of anchors in the RPN part and modified the screening thresholds for foreground and background, the memory consumption and prediction time of a single image are not much different from the original model. From the perspective of comprehensive prediction accuracy and performance, the model processing effect of integrating attention mechanisms and bottom-up FPN is the best. In fact, from the perspective of theoretical analysis, the accuracy of mask RCNN + AM is not much different from that of mask RCNN + FPN. However, in practical application scenarios, those in charge of sampling need to take 50 g (1000 grains) from different parts of a trunk carrying wheat grains as samples for analysis. As a result, we choose the mask RCNN + AM + FPN model as the better result to save time.

In addition, we also recorded the AP value of each kind of unsound kernel, as shown in Table 3. We can see that the AP of each unsound grain in the improved model is higher than the original model, which shows that our variation is valid. Among the three upgraded models, the one with the highest AP value is the one that integrates an attention mechanism and a bottom-up FPN. Among the six kinds of particles, moldy kernels have the highest AP value, and perfect kernels have the lowest. In addition, the result of the model for sprouted and broken kernels is good, and the highest AP values in the four models for spotted kernels and spotted kernels are, respectively, 0.82 and 0.86. However, the model has a poor prediction effect on perfect kernels, with the highest score coming in at 0.6. After analysis, we hold the opinion that the reason is that the characteristics of perfect kernels were not obvious compared with other particles. Therefore, the proportion of the dataset is adjusted later to verify the conjecture. We increased the number of datasets of perfect kernels in different gradients (2×, 3×, 4×, 5×), but the accuracy only increased by 6%, which did not meet our requirements. So far, no reason has been found. 

### 3.3. Comparison with Other Models

In order to further evaluate the recognition efficiency of our model, we compare four network models, including Swin Transformer [26], mask scoring RCNN [27], polar mask [28], and SOLO [29]. They use the same training dataset and validation dataset, and uniformly set the training rounds to 50 epochs. AP, AR, parameters, and time are evaluated by these models. Results are shown in Table 4.

In Table 4, our model achieves the best results in AP and AR. The AR of our model is 86%, which is 28%, 33%, 19%, 69%, and 48% higher than the AR of the other five networks, respectively. Among the six segmentation networks, the polar mask network has the worst performance, with a precision of 17%. In terms of average recall, the average recall rate of our model is 91%, which is 24%, 28%, 18%, 57%, and 38% higher than that of the other five networks, respectively.

The confidence is uniformly set to 0.3, and the prediction results of all models are visualized in Figure 9. In the prediction results of each model, the visualization code is modified to highlight each unsound wheat grain with the same color. Mask scoring RCNN and our prediction results are both better, with relatively few missed detections; under the same threshold, other models miss seriously. However, it is difficult to identify kernels with dense distribution in mask scoring RCNN. SOLO is poor for spotted kernel recognition. The Swin Transformer and polar mask perform poorly on grain edges when segmenting. In fact, when the threshold is set to 0.3, the polar mask does not detect the result. So in order to visualize its segmentation performance, its threshold is set to 0.2. Partially detailed results of the predictions of various models are shown in Figure 10.

## 4. Discussion

### 4.1. Attention Mechanism Module

The most widely used and effective attention mechanisms are SE (Squeeze-and-Energy) [30], ECA (Efficient Channel Attention), and CBAM (Convolutional Block Attention Module) [31]. SE adds attention to the channel dimension. The key operations are squeeze and excitation, which make the model focus on certain feature channels, but SE captures the dependencies of all channels, which is inefficient and unnecessary. ECA is an improvement over SE, which removes the fully connected layer and adds a convolutional layer to give it a good ability to obtain information across channels. CBAM is the combination of channel attention mechanisms and spatial attention mechanisms, and the input feature layers are processed by the channel and spatial attention mechanisms, respectively.

We added these three attention mechanisms, SE, ECA, and CBAM, to the backbone output layers of the mask RCNN model. Figure 11 shows the training loss and mAP. The ECA module has the lowest loss. The model loss after adding the CBAM module and the SE module is not much different, but the model loss after adding the CBAM module is higher. From Table 5, it can be implied that ECA has the highest mAP value and SE has the lowest value, which indicates that the ECA module has the best feature representation ability when only the attention mechanism module is added to mask RCNN. Meanwhile, in terms of memory consumption, the number of model parameters with the ECA module added is small, so the prediction time per image is also the shortest.

### 4.2. FPN Module

In the process of improving the FPN, before adding the bottom-up module, we tried to change the FPN into a recursive FPN (RFPN) [32]. In recursive FPN, we did not use the ASPP [33] module but directly convolved the fused feature map and added it to the corresponding backbone layer. Then, the FPN route of the original model was used for down-sampling and fusion, and finally, 3 × 3 convolution was performed.

However, from the data in Figure 12, although the accuracy of the improved model is improved compared with the original mask RCNN model, its loss is still high, and the accuracy is not greatly improved. Therefore, we find another way by adding a bottom-up module to the FPN part. Meanwhile, the bottom-up FPN module has minimal loss, and its accuracy has greatly improved. Through comparison and analysis, we found that the reason why the result of our improved recursive FPN module is bad is that the operation of convoluting for image resizing and channel expansion lost a lot of information when the image after the first fusion was blended with the output feature map of backbone. Given that the operation of adding ASPP will have an additional impact on the model’s reliability of model, we chose the bottom-up FPN module.

### 4.3. Fusion of Attention Mechanisms and FPN

In Section 4.1 and Section 4.2, we just discussed the model to add or modify a single module without comparing it with the fusion of the two modules above. Therefore, we integrate the three attention modules with the recursive FPN module and the bottom-up FPN module to obtain six fusion methods and compare and analyze the six fusion models with the original model, making a total of seven models. The loss and mAP curves are shown in Figure 13; it can be seen from Figure 13a that the model with the lowest loss is the one with the fusion of an ECA module and a bottom-up FPN module. The loss of the model with the fusion of the ECA module and the RFPN module is next to the above, which is about 10% higher than that. The model after the fusion of the SE module and the bottom-up FPN module has the highest loss, about 38%. Meanwhile, it can be seen from Figure 13b that the model with the lowest loss responds with the highest mAP value, and the model with the highest loss value has the lowest loss value. Therefore, in general, the integrated ECA and bottom-up FPN models have the best identification effect.

Six improved models were obtained by permutations and combinations of three attention mechanisms and two FPN modules, and the values of the evaluation indexes of each model were recorded in detail in Table 6. According to these data, the model with the lowest prediction accuracy is the one integrating SE and a bottom-up FPN module, with an accuracy of only 69%. Meanwhile, this model is also the one with the fewest number of parameters and the shortest time for prediction. The prediction accuracy of the model fused with the SE and RFPN modules is 4% higher than that, and the prediction time is about 2% higher than that. The prediction accuracy of the model fused with the CBAM and RFPN modules is 1% higher, and the time is 0.36 s longer than that of the model fused with the CBAM and bottom-up FPN modules. Overall, there is little difference in the effect of a CBAM module fused with two FPN modules. After the fusion of the ECA and two FPN modules, it is found that the accuracy of the model is greatly improved after the ECA module is fused with the bottom-up FPN module. Meanwhile, the model prediction effect after the fusion of the ECA and RFPN modules is better than that of the other two attention mechanisms fused with the RFPN module, and the time is the lowest. In conclusion, the model with the ECA and bottom-up FPN modules has the highest accuracy and the best performance.

## 5. Conclusions

In this paper, a wheat unsound kernel detection model based on instance segmentation is proposed that could accurately and quickly recognize wheat unsound kernels. When compared to the original mask RCNN and mask scoring RCNN, ours improves accuracy by 28% and 19%, respectively. Instance segmentation estimates each pixel in turn, so it can overcome the problem of grain adhesion that cannot be solved by traditional segmentation. The following are some key conclusions:(1)This model can solve the problem of multi-grain adhesion in dense wheat kernels. It is well known that if a classification network is used, traditional image segmentation methods such as concave segmentation and watershed are needed to solve the adhesion problem, which is not accurate.(2)The mask RCNN network is improved based on the circularity characteristics, the difference between edge features and underlying features such as color and texture, to make it more efficient for multi-target and fine-grained target recognition. Mask RCNN is widely used in the segmentation of the foreground and background, the segmentation of a single class of targets, and the segmentation of the target with a small number of categories. The efficiency is good in the above respects, but for the fine-grained, multi-target segmentation, results are worse. Our model can well overcome the above problems.

However, there are still some drawbacks to our algorithm. On the one hand, the precision of recognition still needs to be improved. On the other hand, the number and quality scores of the unsound kernels need to be further statistically analyzed after the recognition. Therefore, the regression model of area and quality needs to be further constructed to achieve the final calculation of quality content so as to finally evaluate the grade. Finally, in the process of improving the detection accuracy of the perfect kernels, we did not obtain good feedback on increasing the datasets. So, it is necessary to further analyze the reasons for the phenomenon. 

## Figures and Tables

**Figure 1 sensors-23-03379-f001:**
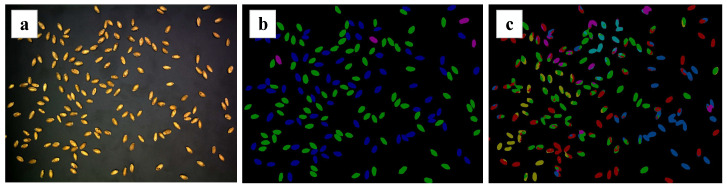
Ground truth and U-Net segmentation results: (**a**) original image; (**b**) ground truth; (**c**) U-Net result.

**Figure 2 sensors-23-03379-f002:**
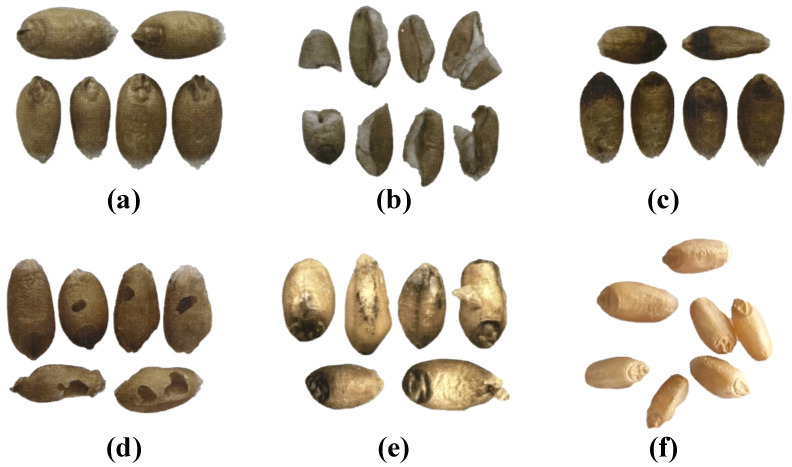
Appearance of unsound kernels and perfect kernels. (**a**) Sprouted kernels; (**b**) broken kernels; (**c**) spotted kernels; (**d**) injured kernels; (**e**) moldy kernels; (**f**) perfect kernels.

**Figure 3 sensors-23-03379-f003:**
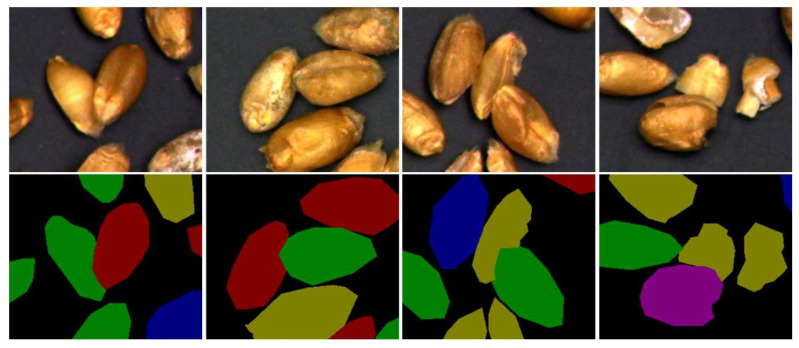
Labeled images of overlapping wheat grain.

**Figure 4 sensors-23-03379-f004:**
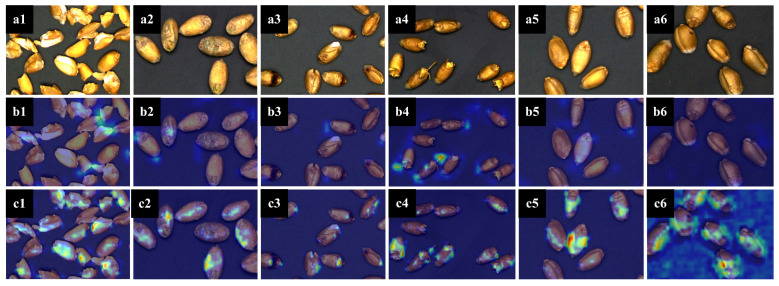
Heat-MAP of unsound kernels and perfect kernels. (**a1**) Original image of broken kernels. (**a2**) Original image of moldy kernels. (**a3**) Original image of spotted kernels. (**a4**) Original image of sprouted kernels. (**a5**) Original image of perfect kernels. (**a6**) Original image of injured kernels. (**b1**) Mask RCNN heat-MAP of broken kernels. (**b2**) Mask RCNN heat-MAP of moldy kernels. (**b3**) Mask RCNN heat-MAP of spotted kernels. (**b4**) Mask RCNN heat-MAP of sprouted kernels. (**b5**) Mask RCNN heat-MAP of perfect kernels. (**b6**) Mask RCNN heat-MAP of injured kernels. (**c1**) Improved mask RCNN heat-MAP of broken kernels. (**c2**) Improved mask RCNN heat-MAP of moldy kernels. (**c3**) Improved mask RCNN heat-MAP of spotted kernels. (**c4**) Improved mask RCNN heat-MAP of sprouted kernels. (**c5**) Improved mask RCNN heat-MAP of perfect kernels. (**c6**) Improved Mask RCNN heat-MAP of injured kernels.

**Figure 5 sensors-23-03379-f005:**
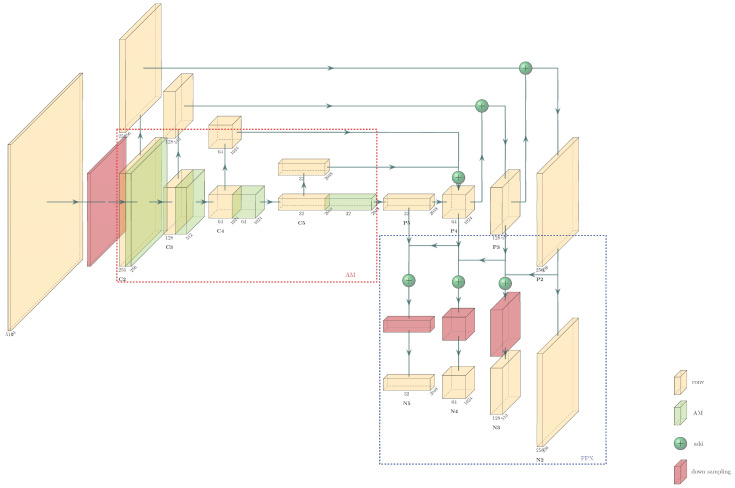
Structure of improved FPN.

**Figure 6 sensors-23-03379-f006:**
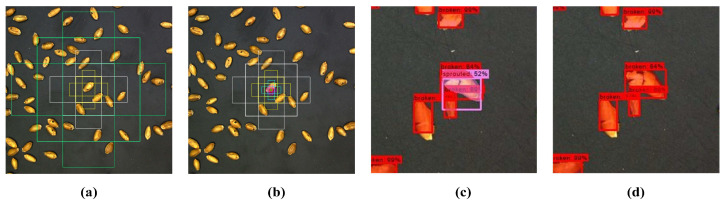
Comparison of RPN before and after lifting. (**a**) Anchor boxes before adjustment. (**b**) Anchor boxes after adjustment. (**c**) Image result before adjusting the NMS threshold. (**d**) Image result after adjusting the NMS threshold.

**Figure 7 sensors-23-03379-f007:**
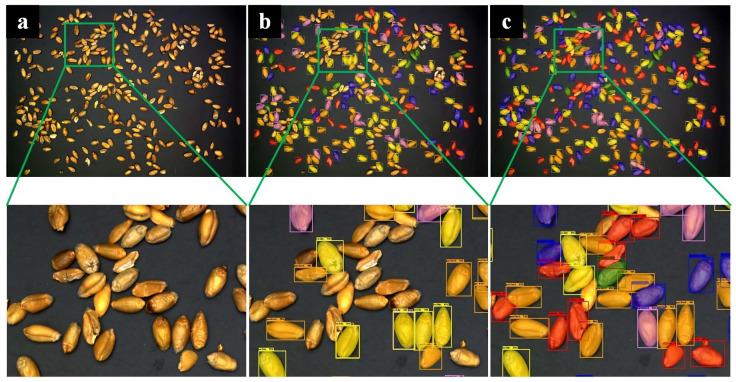
Result images. (**a**) Original image. (**b**) Result images of the mask RCNN model. (**c**) Result images of the improved mask RCNN model (representations of color boxes are as follows: orange: perfect kernels; yellow: moldy kernels; red: broken kernels; purple: sprouted kernels; blue: spotted kernels; green: injured kernels).

**Figure 8 sensors-23-03379-f008:**
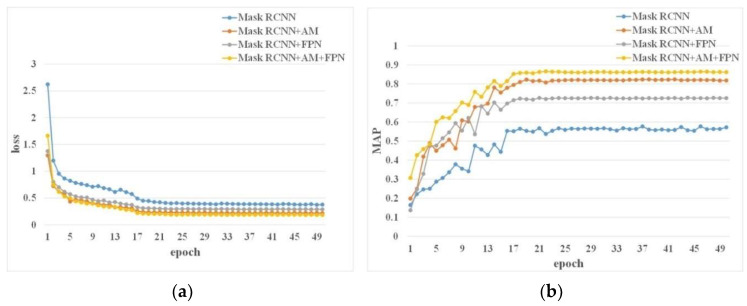
Loss and mAP of the models. (**a**) Loss curves of the models. (**b**) mAP curves of the models.

**Figure 9 sensors-23-03379-f009:**
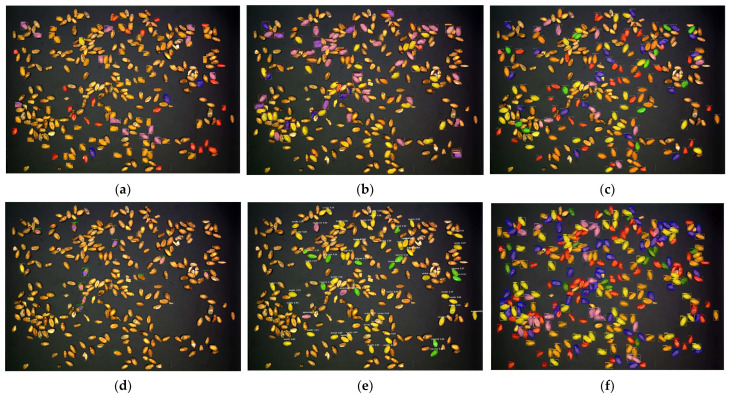
Result images. (**a**) Original image. (**b**) Result images of Swin Transformer. (**c**) Result images of mask scoring RCNN. (**d**) Result images of polar mask. (**e**) Result images of SOLO. (**f**) Result images of ours (representations of color boxes are as follows: orange: perfect kernels; yellow: moldy kernels; red: broken kernels; purple: sprouted kernels; blue: spotted kernels; green: injured kernels).

**Figure 10 sensors-23-03379-f010:**
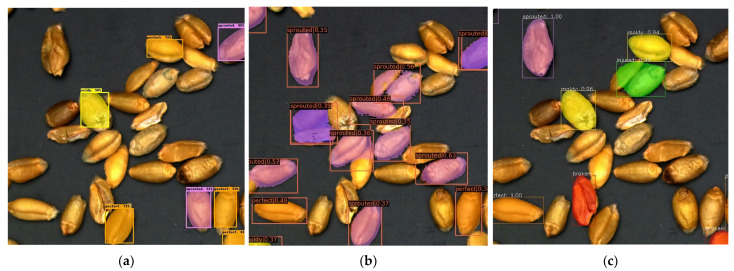
Result images with detailed information. (**a**) Original image. (**b**) Result images of Swin Transformer. (**c**) Result images of mask scoring RCNN. (**d**) Result images of polar mask. (**e**) Result images of SOLO. (**f**) Result images of ours (representations of color boxes are as follows: orange: perfect kernels; yellow: moldy kernels; red: broken kernels; purple: sprouted kernels; blue: spotted kernels; green: injured kernels).

**Figure 11 sensors-23-03379-f011:**
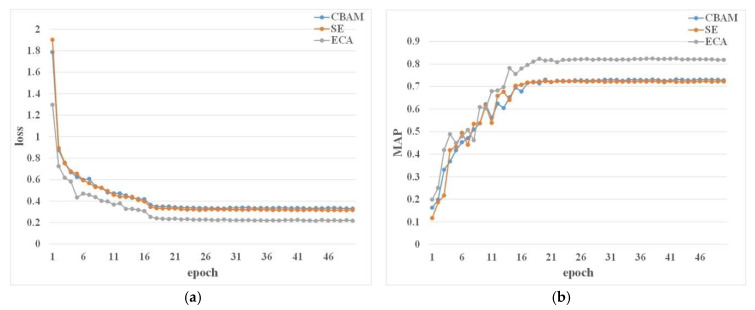
Loss and mAP of the models with different AM modules. (**a**) Loss of the models with different AM modules. (**b**) mAP of the models with different AM modules.

**Figure 12 sensors-23-03379-f012:**
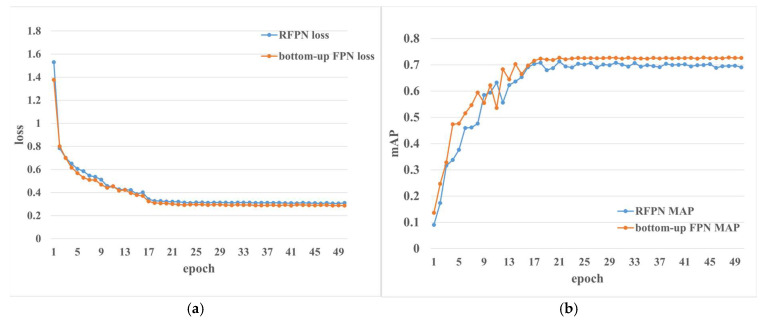
Loss curves and mAP curves after adding improved FPN module. (**a**) Loss of the models with different FPN modules. (**b**) mAP of the models with different FPN modules.

**Figure 13 sensors-23-03379-f013:**
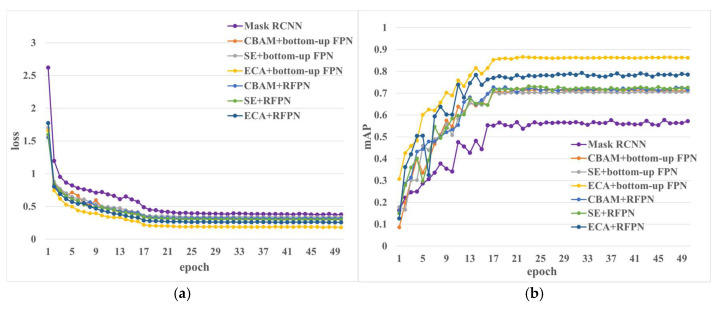
Loss and mAP of the models with different AM and FPN modules. (**a**) Loss of the model with fusing AM and FPN. (**b**) mAP of the model with fusing AM and FPN.

**Table 1 sensors-23-03379-t001:** Distribution of the dataset.

Kernel Types	Number of Training Images	Number of Validation Images	Number of Test Images
Perfect	24	3	3
Broken	24	3	3
Moldy	24	3	3
Spotted	24	3	3
Sprouted	24	3	3
Injured	24	3	3
Mixture	480	60	60
Total	624	78	78

**Table 2 sensors-23-03379-t002:** Validation metrics of different models.

Module	Precision	Recall	Parameters	Time/s
Mask RCNN	0.58	0.67	9203	6.18
Mask RCNN + AM	0.82	0.89	9521	7.10
Mask RCNN + FPN	0.72	0.76	9318	6.67
Mask RCNN + AM + FPN	0.86	0.91	9530	7.81

**Table 3 sensors-23-03379-t003:** AP values of different kernels under different models in validation.

Model	mAP	AP					
		Perfect	Moldy	Injured	Spotted	Sprouted	Broken
Mask RCNN	0.58	0.42	0.74	0.46	0.42	0.73	0.69
Mask RCNN + AM	0.82	0.53	0.97	0.75	0.76	0.91	0.98
Mask RCNN + FPN	0.72	0.46	0.91	0.69	0.83	0.83	0.89
Mask RCNN + AM + FPN	0.86	0.60	0.99	0.82	0.89	0.89	0.98

**Table 4 sensors-23-03379-t004:** Comparison of six segmentation methods in validation.

Module	AP	AR	mIOU	Parameters	Time/s
Mask RCNN	0.58	0.67	0.60	9203	6.18
Swin Transformer	0.53	0.63	0.53	6513	5.10
Mask Scoring RCNN	0.67	0.73	0.77	9210	6.67
Polar Mask	0.17	0.34	0.25	7653	5.61
SOLO	0.38	0.53	0.44	8125	6.12
Improved Mask RCNN	0.86	0.91	0.87	9530	7.81

**Table 5 sensors-23-03379-t005:** Evaluation results of three attention modules in validation.

Module	Precision	Recall	Parameters	Time/s
SE	0.72	0.76	9528	7.80
CBAM	0.73	0.78	9538	7.96
ECA	0.82	0.89	9521	7.10

**Table 6 sensors-23-03379-t006:** The evaluation values of the six models in validation.

Model	Precision	Recall	Parameters	Time/s
SE + RFPN	0.73	0.78	9545	8.12
SE + bottom-up FPN	0.69	0.74	9538	7.96
CBAM + RFPN	0.72	0.75	9560	8.27
CBAM + bottom-up FPN	0.71	0.76	9540	7.91
ECA + RFPN	0.78	0.81	9538	7.96
ECA + bottom-up FPN	0.86	0.91	9530	7.83

## Data Availability

Our datasets were analyzed in this study. These data can be found by shenranVic/wheat-detection (github.com, accessed on 1 December 2022).

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
