# Peer review of "Segmentation of Unsound Wheat Kernels Based on Improved Mask RCNN"

_sensors, 2023, doi:10.3390/s23073379_

Round 1
Reviewer 1 Report
My comments and suggestions are stated in the attached file.

Reviewer 2 Report
Grammar throughout the manuscript must be improved. Many sentences contain multiple dissimilar thoughts which makes it difficult to understand the ideas the authors are trying to convey. These are some of the thoughts I recorded while reading the manuscript:
1. Introduction: Grammar needs to be improved. For clarity, many of the long sentences need to be simplified (example sentences: lines 32-34, 51-54, 67-71)
Line 37: Incomplete paragraph.
2. Materials and Methods
Line 111 – 112: It is difficult to understand what information the authors are trying to convey in this sentence. If broken and moldy kernels are considered separately, then use different sentences to describe them.
Line 123: spell 130. (spell numbers if they begin a sentence)
Lines 124 – 127: sentence needs to be simplified and clarified.
Lines 127 – 131: sentence needs to be simplified and clarified.
Lines 147 – 154: uncertain what idea the authors are trying to convey; sentence needs to be simplified and clarified.
Lines 154 – 158: simplify
Line 184: closed parenthesis missing
Line 191: windows not Windows
3. Results
Lines 248 – 253: an example of a run-on sentence that needs to be simplified for clarification; try rewriting as multiple sentences
Line 306: therefore not Therefore
Present differences as a percentage (example: line 272, 319, 322, etc)
Reviewer 3 Report
In this manuscript, an improved Mask RCNN-based algorithm for identifying non-sound seeds of wheat is proposed. Experimental results show that the improved Mask RCNN algorithm can identify unhealthy kernels faster and more accurately, and can handle the adhesion problem well. And it has better accuracy and performance compared with other segmentation models. However, there are still some issues should be addressed.
1. In section 2.1.1, is there a clearer image to show the appearance of the various types of wheat grains?
2. In section 2.2.3 indicates improvements to the RPN and shows that, to some extent, these methods significantly reduce computational effort and improve detection efficiency, and whether there are actual results to demonstrate the effectiveness of this improved method.
3. The evaluation metrics of common segmentation algorithms are described in Section 2.3, but the evaluation metrics used in this paper in addition to time are not specified.
4. In the quantitative results of Section 3.2, there is not much difference in the combined prediction accuracy and performance of Mask RCNN+AM+FPN model compared with Mask RCNN+AM+FPN model, and the computing time is also faster than Mask RCNN+AM+FPN model, so why not use this model, please conduct a more detailed comparison to explain.
5. In section 3.2 mentions that the Mask RCNN+AM+FPN model has poor prediction for perfect kernels, and later adjusts the dataset scale to verify this conjecture, do you show the adjusted results and the conclusion in the later paper?
6. In section 4.3, it is mentioned that the six fusion models are compared with the original model for analysis, but only the results of the six fusion models are shown in Figure 12, and it is suggested that the results of the original model be added as well, so that the comparative analysis will be more intuitive.
